# Semi-Empirical Satellite-to-Ground Quantum Key Distribution Model for Realistic Receivers

**DOI:** 10.3390/e25040670

**Published:** 2023-04-17

**Authors:** Aleksandr V. Khmelev, Egor I. Ivchenko, Alexander V. Miller, Alexey V. Duplinsky, Vladimir L. Kurochkin, Yury V. Kurochkin

**Affiliations:** 1Landau Phystech School of Physics and Research, Moscow Institute of Physics and Technology, Dolgoprudny, Moscow 141701, Russia; 2Russian Quantum Center, Skolkovo, Moscow 143025, Russia; 3QSpace Technologies, Moscow 121205, Russia; 4Moscow Centre for Quantum Technologies, Moscow 127204, Russia; 5Moscow Institute of Electronics and Mathematics, HSE University, Moscow 101000, Russia; 6NTI Center for Quantum Communications, NUST MISiS, Moscow 115419, Russia; 7Quantum Research Centre, Technology Innovation Institute, Masdar City, Abu Dhabi 9639, United Arab Emirates

**Keywords:** satellite quantum communication, quantum key distribution, optical communication, free-space optics, quantum channel modeling

## Abstract

Satellite-based link analysis is valuable for efficient and secure quantum communication, despite seasonal limits and restrictions on transmission times. A semi-empirical quantum key distribution model for satellite-based systems was proposed that simplifies simulations of communication links. Unlike other theoretical models, our approach was based on the experimentally-determined atmospheric extinction coefficient typical for mid-latitude ground stations. The parameter was measured for both clear and foggy conditions, and it was validated using published experimental data from the Micius satellite. Using this model, we simulated secure QKD between the Micius satellite and ground stations with 300 mm and 600 mm aperture telescopes.

## 1. Introduction

Quantum communication technology provides a theoretically unhackable communication channel for encryption key exchange and long-term data security [1]. Based on quantum physics principles, the method, also known as a quantum key distribution (QKD), allows users to detect eavesdropping and estimate key leakage caused by hacker-induced errors in quantum states [2,3].

Typically, quantum communications rely on photon states being transferred across an optical fiber or through the atmosphere. However, the amount of light in the medium decreases exponentially with distance, and no replication or amplification is possible. As a result, the feasible range of QKD in optical fibers [4,5] and in terrestrial free space [6,7] is limited to a few hundred kilometers. Satellite-based QKD was proposed to expand the range of quantum communication because most of the signal transmission path between the satellite and the Earth passes in a vacuum. Several scientific organizations initiated satellite-based QKD research projects and demonstrated certain technological concepts [8,9].

Significant accomplishments were reported in Japan, where results on the conservation polarization characteristics of the laser beam from a satellite and a link parameters estimation for satellite-to-ground QKD were provided [10]. The first quantum-limited experiment between a low-Earth orbit (LEO) satellite and a ground station was conducted [11]. Quantum-limited tests were carried out on a geostationary satellite in 2017 [12]. In 2016, researchers from USTC (China) launched Micius, the first operating quantum science satellite, and successfully demonstrated satellite-to-ground QKD via polarization photon states [13,14].

As a promising concept, the first quantum science satellite paved the way for a future global quantum network, enabling secure worldwide data transfer [15,16,17]. Meanwhile, as a communication network capable of providing secret keys as a service, industrial quantum key distribution networks must meet the core requirements of flexible extension, cost effectiveness, component compatibility, and standardization [18,19,20,21]. As a result, a strict and comprehensive analysis of a quantum communication channel is needed to build an efficient quantum network [22,23], despite seasonal limits and restrictions on transmission times between satellites and ground stations.

Satellite-based uplink–downlink [24] and measurement-device-independent QKD models [25] were proposed recently. Additionally, a theoretical analysis of the quantum communication link between the Micius satellite and the Delingha receiving ground station was performed [22,26]. The existing theoretical models are valuable for a comprehensive analysis of free-space QKD. However, these models take into consideration many factors not known with sufficient accuracy, such as air turbulence, background noise, and pointing and tracking losses [8], which significantly limit their practical usage.

In this paper, we present a semi-empirical satellite-to-ground QKD model relying on phenomenological data by employing atmospheric extinction measurements. This allows for a realistic analysis of developed ground stations with 300 and 600 mm telescope apertures in terms of secure key rate and error rate. Considering our simulation of experimental results obtained with Micius [13] and a theoretical model [8,22], a more relevant and real range of sifted key rate for the satellite-based QKD was provided.

This paper is organized as follows: in Section 2, we describe the communication channel geometry for a series of satellite passages over Zvenigorod observatory and computed the time-dependent distance between the satellite and ground stations; in Section 3, we calculated overall link losses based on the measured characteristics of ground stations [27] and the photon source parameters of a satellite [13]; Section 4 examined the secure key generation and error rates of QKD systems using the standard BB84 decoy-state protocol [28,29]; finally, in Section 5, we discuss the deployment of our model to future space-to-ground quantum communication experiments.

## 2. Satellite-to-Ground Communication Geometry

To determine the channel efficiency in a satellite-based QKD model, one needs to calculate the time-dependent distance d(t) between the transmitter and the receivers, as well as the satellite elevation angle $\theta_{El}$ above the horizon. We considered several typical satellite passages and calculated one passing through the zenith in a circular orbit.

Three non-zenith actual passages of the Micius satellite were simulated for 31 October 2021, 9 March 2022, and 23 March 2022, with the maximum elevation angles of 32.5°, 57.5°, and 42°, respectively. To compute the time dependency of the satellite’s elevation angle for the (55°$42^{\prime }\;$N, 36°$45^{\prime }\;$E) location, we used the simplified general perturbations model SGP4 [30], given the satellite’s two-line elements (TLE) orbital data [31,32].

We also considered a simple model to calculate the trace-time function of the elevation angle for the limit case corresponding to the satellite passing through the zenith point over the ground station.Here, we assumed that the Micius satellite moves in a perfectly circular orbit (the eccentricity is 0) at an altitude of 500 km and an inclination of 97.33° [13].

The angular velocity ω of a low-Earth orbit satellite may be approximated by a constant in the Earth-centered fixed frame, according to Ref. [33]: (1)$$\omega \approx \omega_{S}-\omega_{E}\cos i\;,$$
where $\omega_{S}\approx \sqrt{g/R}$ is the angular velocity of the satellite in the Earth-centred inertial coordinate system with acceleration due to gravity of *g* and an orbit radius of *R*, the orbit inclination is *i*, and $\omega_{E}$ is the angular velocity of the Earth’s rotation. The numerical values of the orbital parameters and angular velocity data of the Micius satellite for our model are listed in Table 1.

We approximated the magnitude of the tangential velocity by its minimum value for a given orbit in the Earth-centered fixed frame. The absolute fluctuation of the tangential velocity magnitude for the LEO satellite is quite negligible for any latitude, according to Ref. [34]. Therefore, angular velocity is independent of the ground station latitude in this model.

The elevation angle for the zenith satellite passage was calculated using a set of equations derived from simple geometric considerations (see Figure 1 and Section A.1), as follows: (2)$$\theta_{El}=\arccos\left(\sqrt{\frac{\sin^{2}\alpha }{1+{\left(\frac{R_{E}}{R}\right)}^{2}-2\left(\frac{R_{E}}{R}\right)\cos\alpha }}\right),$$
where the central angle can be expressed as a linear function of time as a α=ωt.

Next, we determined the distance between a satellite and a ground station using the elevation angle from TLE data over time for real satellite passage as well as a computed satellite pass above the zenith. Section A.2 has a detailed derivation of the distance, which is given by: (3)$$d^{2}+d\cdot 2R_{E}\sin\theta_{El}+R_{E}^{2}-R^{2}=0.$$

Figure 2 shows the dependence of the satellite elevation angle and the distance between the satellite and the ground station on time obtained from Equations (Equation 1)–(Equation 3) for the typical satellite passages, by using TLE data and the calculated zenith satellite passage. The calculations were carried out when the satellite was in the ground station’s field of view, i.e., when the elevation angle was above 20 degrees above the horizon.

The time duration of quantum communication increases with a peak elevation angle in Figure 2, whereas the minimal distance between the satellite and receiver decreases. The satellite passage through the zenith offers the longest communication time and and the shortest communication channel length. For comparison, the communication time of four satellite passages with the maximum elevation angles of 32.5°, 42°, 57.5°, and 90° are equal to 221 s, 260 s, 283 s, and 285 s, respectively.

## 3. Link Efficiency for 300-mm and 600-mm Aperture Ground Stations

Channel losses between a satellite and a ground station dynamically change during quantum communication, which significantly distinguishes this method from fiber optic or terrestrial QKD. In general, diffraction losses vary with the change of the distance between a satellite and a ground station. Additionally, varying the elevation angle results in various effective thicknesses of the atmosphere during light transmission.

The diffraction loss can be calculated using the laser source divergence γ, communication channel length *d*, and effective receiving area with telescope diameter $D_{T}$ and obstruction ε of the telescope secondary mirror, as follows $\varepsilon D_{T}^{2}/{\left(\gamma d\right)}^{2}$. However, the atmospheric extinction should be described in more detail.

In astronomy, the magnitude of a star or another visible celestial object, such as a satellite, indicates its brightness class. The magnitudes $m_{1}$ and $m_{2}$ of two objects are related to the corresponding flux densities $F_{1}$ and $F_{2}$ by equation [35]: (4)$$m_{1}-m_{2}=-2.5\log_{10}\left(F_{1}/F_{2}\right).$$

Given that starlight travels a greater distance through the atmosphere near the horizon than it does at the zenith, it follows that a star near the horizon will be less luminous. Therefore, we can relate the magnitude of the observed object outside the atmosphere $m_{0}$ (apparent magnitude) to the magnitude of the object at the Earth’s surface *m*, as follows: (5)$$m-m_{0}=\varkappa \cdot f\left(\theta_{El}\right),$$
where ϰ is the atmospheric extinction coefficient and $f(\theta_{El})$ is the air mass function depending on the elevation angle of the object. The amount of air at the zenith corresponds to one air mass.

Young and Irvine [36] proposed the relationship between the air mass function and elevation angles as: (6)$$f\left(\theta_{El}\right)=\csc\theta_{El}\cdot \left(1-0.0012\cot^{2}\theta_{El}\right).$$

The atmospheric extinction coefficient was determined from the measurements of stars’ fluxes with known apparent magnitudes outside the atmosphere in the spectral range 845 nm–855 nm [37] at several elevation angles by means of a 600-mm ground station. We assumed that, for the same ground station, the star fluxes were equally proportional to the photon count rate. Appendix B contains additional information and data.

Figure 3 depicts the dependence of atmospheric extinction $\varkappa f(\theta_{El})$ on air mass up to a constant, according to Equations (Equation 25) and (Equation 6). The measurements of star fluxes were performed on 9 March 2022, with clear weather conditions (humidity <20%) and on 24 June 2021, with hazy, foggy weather conditions (humidity <80% near the Earth’s surface). As a result, the slope of atmospheric extinction equals the atmospheric extinction coefficient and ranges from 0.23±0.08 for clear nights to 0.41±0.09 for foggy nights.

Hence, link efficiency, or the overall transmission of photons at 850 nm between the Micius satellite and a ground station, including detection efficiency, is given by: (7)$$\eta \left(t\right)=\frac{\varepsilon D_{T}^{2}}{{\left(\gamma d\right)}^{2}}\cdot 10^{-0.4\varkappa \csc\theta_{El}\cdot \left(1-0.0012\cot^{2}\theta_{El}\right)}\eta_{opt}\eta_{det},$$
where $\eta_{opt}$ is an optical efficiency of the respective ground station and $\eta_{det}$ is a quantum efficiency of a single photon detector.

Table 2 summarizes the baseline parameters of the setups required for further modeling. The 600 mm ground station is a fixed telescope equipped with a quantum states decoder, as detailed in Ref. [27]. The 300 mm ground station is a portable receiving telescope equipped with a quantum communications optical system [38], which has a better optical efficiency than a 600 mm telescope. The detailed description of the experimental setups are provided in Appendix C.

Figure 4 presents calculated link loss $\eta_{link}=-10\log_{10}\eta (t)$ for our satellite passages in the satellite QKD experiment under clear weather conditions. For the given system parameters, the peak link losses at zenith for the ground stations with the telescope apertures of 300 mm and 600 mm are equal to 32 dB and 29 dB, respectively. The characteristic difference of channel losses between elevation angles of 20° and 90° are 9 dB for both receivers.

## 4. Satellite-to-Ground QKD Analysis

In this section, we simulated a satellite-to-ground QKD experiment using the BB84 decoy-state protocol [28,39]. We determined the sifted key rate $R_{sift}$ and quantum bit error rate (QBER) of the designed receivers. Then, we estimated the secret key length by applying data post-processing for signal state QKD.

### 4.1. Satellite-QKD Model and Parameters Verification

We used the QKD model from Ref. [28], with the difference that the link transmission was time-dependent. Therefore, transmission of a *k*–photon state through the communication channel described in Section 3 is given by: (8)$$\eta_{k}=1-{\left(1-\eta \left(t\right)\right)}^{k}$$
for k=0,1,2,⋯.

The photon number in each weak coherent pulse of the satellite transmitter has a Poisson distribution [13]. Pulses with a frequency *f* and intensities of signal μ, decoy ν, and vacuum λ=0 states are emitted at random with probabilities $p_{s}$, $p_{d}$, and $p_{v}$, respectively.

Background noise $Y_{0}$, which includes detector dark counts as well as additional background contributions, was considered to be 250 and 500 counts per second for ground stations with 300 mm and 600 mm apertures, respectively. Note that the values were assumed to be constant during communication and were obtained while tracking the switched-off satellite by the ground stations.

Additionally, intrinsic error $e_{det}$ caused by receiver measurement fidelity remained independent of distance and was set at 0.9% and 0.5% for the 300 mm and 600 mm ground stations, respectively. The values used to simulate our QKD setups are listed in Table 3.

The sifted key rate $R_{sift}$ and QBER were calculated from parameters of the optical link for the weak and vacuum decoy-state BB84 model provided by Xiongfeng Ma et al. [28]. The expressions of values for signal states are given by: (9)$$R_{sift}=\frac{1}{2}fp_{s}\left[Y_{0}+1-e^{-\mu \eta (t)}\right],$$
(10)$$QBER=\frac{1}{2}fp_{s}\left[e_{0}Y_{0}+e_{det}\left(1-e^{-\mu \eta (t)}\right)\right]/R_{sift}.$$

Figure 5 presents the calculated $R_{sift}$ and QBER of signal states during the satellite passages with peak elevation angles of 32.5° and 90°. The sifted key rate has a strong dependence on the maximum elevation angle and changes non-uniformly over the communication time.

We list the sifted key rate at a maximum value are 3.3 kbit s${}^{-1}$ and 11.4 kbit s${}^{-1}$ for the 300 mm ground station, and 7.2 kbit s${}^{-1}$ and 25.3 kbit s${}^{-1}$ for the 600 mm one. The minimal sifted key generation corresponds to the elevation angle of 20° and equals 1.4 kbit s${}^{-1}$ and 3.1 kbit s${}^{-1}$ for the ground stations with the 300 mm and 600 mm apertures, respectively. The QBER of signal states $E_{\mu }$ is also different for the ground stations [28] and is limited to 3% for the 300 mm aperture ground station, and to 2.5% for the 600 mm aperture.

To verify our semi-empirical model with a phenomenological parameter ϰ, we compared the simulated results for the Xinglong ground station with published experimental data collected with the Micius satellite [13] and modelled findings by Vergoossen et al. [22]. Figure 6 illustrates the peak sifted key rate on the shortest distance between the Micius satellite and the Xinglong ground station for specific satellite passages between September 2016 and May 2017.

The Xinglong ground station is equipped with a 1000-mm-aperture telescope, where the obstruction of the telescope’s secondary mirror is approximated to be 80%. In order to compare the models, we calculated the atmospheric extinction coefficient $\varkappa_{Vergoossen}=0.80$, which is equivalent to a −3.2 dB Vergoossen atmospheric absorption loss at zenith. The other inputs for the key rate calculations are provided in Vergoossen et al. (Table 3 [22]).

Thus, the peak sifted key rate in our model with the atmospheric extinction parameter $\varkappa_{Vergoossen}$ agrees well with the model prediction provided by Vergoossen et al. [22]. Meanwhile, the experimentally obtained atmospheric extinction coefficient for mid-latitude ground stations under favourable weather conditions provides a better evaluation of the Micius data with the shortest distances of 530.33 km, 591.56 km, 711.94 km, and 929.67 km. Moreover, the atmospheric extinction coefficient for foggy, hazy weather conditions, together with less than 3 dB loss due to pointing error [13], is suitable for the assessment of the lower bound of the peak sifted key rate.

Hence, we have verified our model with the real experimental data. As a result, our semi-empirical model with phenomenological parameters is robust and closely matches the satellite-to-ground QKD experiment.

### 4.2. Secret Keys for 300 mm and 600 mm Ground Stations

We stress that only single photon states are completely secure in practical QKD with an imperfect single photon source [40]. The communication time of Δt corresponds to elevation angles greater than 20° and sifted key length $l_{sift}$ is given: (11)$$l_{sift}=\int_{\Delta t}R_{sift}\;dt.$$

Therefore, we estimated the proportion of secret bits from obtained sifted keys using the classic BB84 decoy-state protocol, according to Refs. [28,39]. The lower limit of the final secret key length $l_{sec}$ is as follows: (12)$$l_{sec}\geq Np_{s}\;q\left\{Q_{1}\left[1-H_{2}\left(e_{1}\right)\right]-1.44Q_{\mu }H_{2}\left(E_{\mu }\right)\right\},$$
where *N* is the overall number of pulses emitted by satellite, q=1/2 is the BB84 protocol due to sifting procedure, the subscript μ denotes the intensity of signal states, $Q_{\mu }$ is the gain of signal states, $E_{\mu }$ is the mean overall QBER across communication time, $Q_{1}$ is the gain of single photon states, $e_{1}$ is the error rate of single photon states, the $H_{2}\left(x\right)$ is the binary Shannon entropy function, and 1.44 is the efficiency of error correction [41].

Given the finite size of the sifted key length, the average values contain statistical fluctuations that can be estimated via Chernoff inequality [42] and the observed values. The failure probability for the Chernoff bound is equal to $10^{-9}$.

Table 4 shows the calculated sifted key length and the estimated secret key length, using Equation (Equation 12) and Chernoff inequality, for four satellite passages with the maximum elevation angles of 32.5°, 42°, 57.5°, and 90°.

For typical satellite passages with various peak elevation angles, the secret key lengths differ by 3.2 to 4.6 times for the ground stations with 300 mm and 600 mm apertures, whereas telescope areas differ by four times. For some satellite passages, the 300 mm ground station is more cost-effective and efficient for space-based quantum communications.

## 5. Conclusions

A semi-empirical QKD model for satellite-based systems that simplifies the practical simulation of communication links has been proposed. In contrast to other pure theoretical models, our semi-empirical approach was based on atmospheric extinction coefficients common for ground stations at mid-latitude. These coefficients were determined experimentally by means of real ground-based receivers developed by the authors.

To verify our method, the Xinglong ground station’s peak sifted key rate was simulated and compared to the published experimental data from the Micius satellite for specific satellite passages. The lower and upper boundaries of the peak sifted key rate were appropriately estimated by our model, employing best-choice phenomenological parameters. Then, using this model for ground stations with 300 and 600 mm aperture telescopes, we simulated the secure key generation from the Micius satellite.

To conclude, we proposed and verified a simple practical method for estimating the performance of ground-based stations, which are one of the key elements of future QKD networks. We believe that the developed model will be useful for quick, right-on-the-spot estimation of satellite QKD links and the analysis of future satellite-to-ground QKD systems.

## Figures and Tables

**Figure 1 entropy-25-00670-f001:**
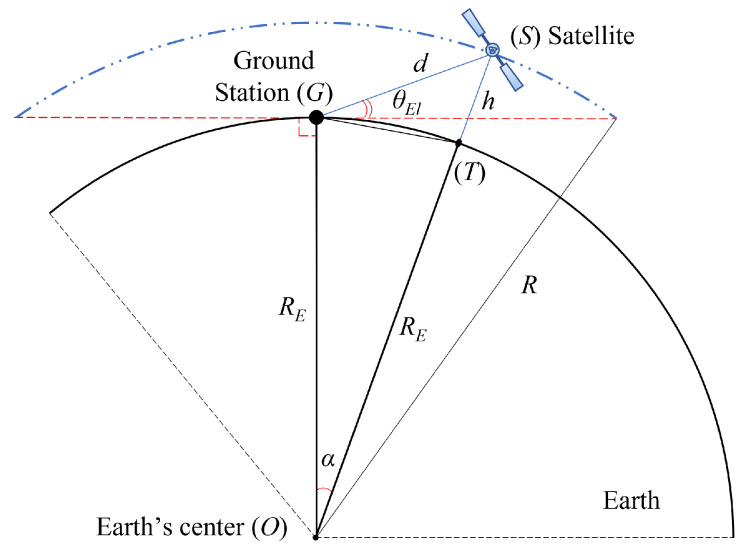
Schematic of the Earth-satellite geometry. The satellite passing over the ground station through the zenith (i.e., $\theta_{El}^{max}=90$°) has an orbit radius of $R=R_{E}+h$, where $R_{E}$ is the Earth’s radius (6364 km) for the Zvenigorod location and *h* is the altitude of the satellite orbit. The distance between the satellite and the ground station is determined solely by an elevation angle $\theta_{El}$, that is, the angle between the horizontal line of sight and the direction of the satellite from the site of the ground station. The central angle between a direction on the ground station and a direction on the satellite from the Earth’s center is denoted by α.

**Figure 2 entropy-25-00670-f002:**
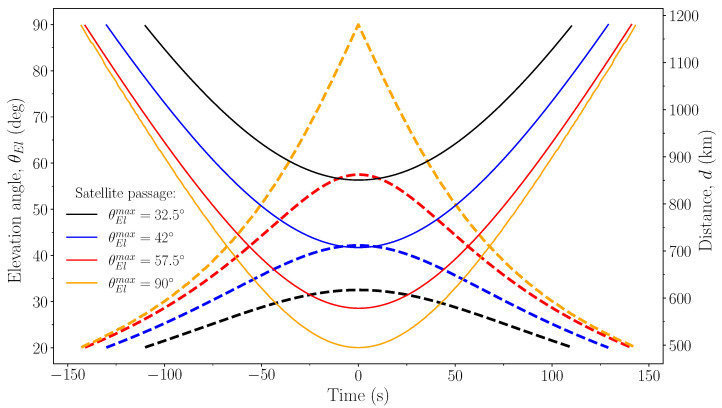
Calculated characteristics of the satellite passages. Dashed curves: elevation angle for the actual passages using the satellite TLE data and for the limit zenith passage with a maximum elevation angle of 90°. Solid curves: communication channel length for the actual passages and for the satellite passage through the zenith. The satellite passages are symmetrically aligned with respect to the highest elevation angle in each one that corresponds to zero time.

**Figure 3 entropy-25-00670-f003:**
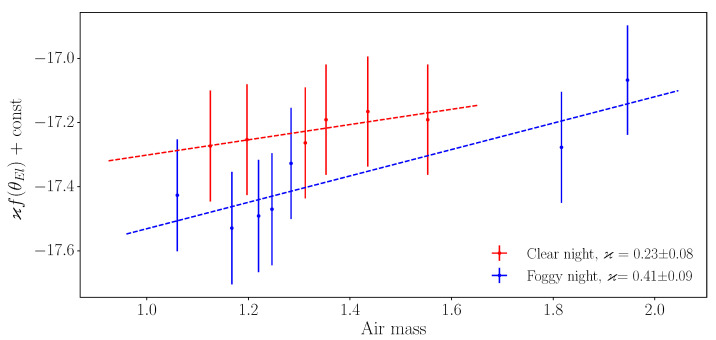
The experimental data and the trend lines of the relationship between atmospheric extinction increase and air mass as estimated by stars’ fluxes observations on clear and foggy nights. The raw data are listed in Table A1 and were acquired with the 600-mm ground station.

**Figure 4 entropy-25-00670-f004:**
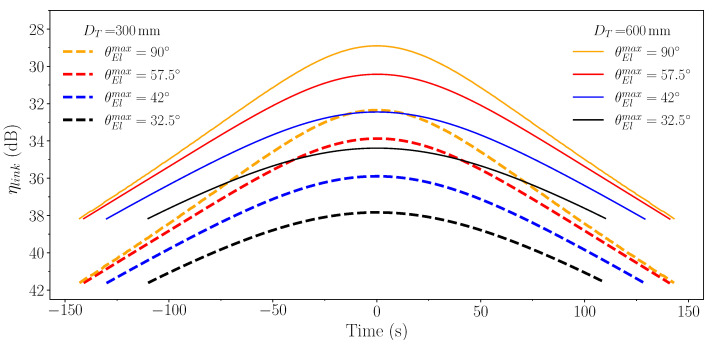
Link losses calculated for real passes of the Micius satellite over Zvenigorod observatory, with max elevation angles of 32.5° (31 October 2021; black curves), 57.5° (9 March 2022; red curves) and 42° (23 March 2022; blue curves). The yellow curves represent theoretical zenith passage with a maximum elevation angle of 90°. Solid lines exhibit the 600 mm telescope data; dashed lines—300 mm. The computations are carried out for elevation angles greater than 20°. We assumed good weather conditions, that is, ϰ=0.23.

**Figure 5 entropy-25-00670-f005:**
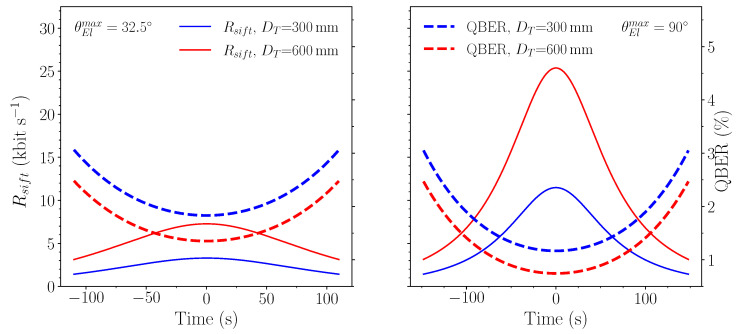
Modelled sifted key rate $R_{sift}$ and QBER of signal states during the satellite passage above a ground station on 31 October 2021 and during the zenith satellite passage. The key parameters of the model are listed in Table 3. The computations were carried out for clear weather conditions and for elevation angles greater than 20°.

**Figure 6 entropy-25-00670-f006:**
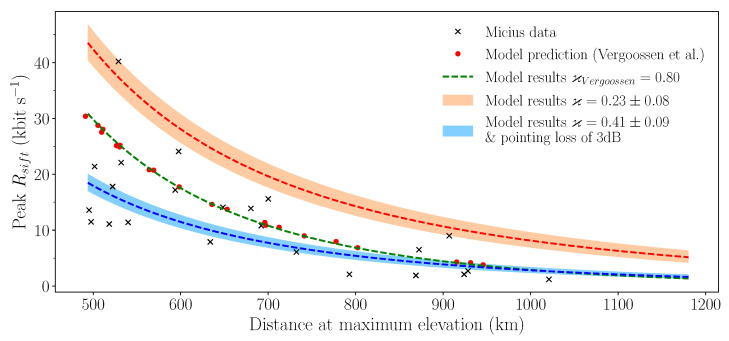
Comparison of peak sifted key rates simulated by a semi-empirical model with Vergoossen’s model data as a function of the shortest distance between satellite and ground stations. The experimental data collected by the Micius satellite for specific satellite passages are presented [13].

**Table 1 entropy-25-00670-t001:** The orbital parameters of the satellite.

*R* (km)	*i* (Degrees)	$\omega_{E}$ (Rad/s)	$\omega_{S}$ (Rad/s)
6864	97.3	$7.3\times 10^{-5}$	$111.4\times 10^{-5}$

**Table 2 entropy-25-00670-t002:** Experimental setup parameters for modeling channel between satellite and ground stations [27,37,38].

$D_{T}$ (mm)	ε	$\eta_{\mathrm{opt}}$	$\eta_{\det}$	γ (Rad)	ϰ
300	0.81	0.44	0.55	$10^{-5}$	0.23–0.41
600	0.73	0.27	0.55	$10^{-5}$	0.23–0.41

**Table 3 entropy-25-00670-t003:** Key parameters of the QKD experimental setups.

μ	$p_{s}$	***f* (Hz)**	$Y_{0}^{300}$ **(/pulse)**	$e_{\det}^{300}$
0.8	0.5	$10^{8}$	$2.5\times 10^{-6}$	0.9%
ν	$p_{d}$	$p_{v}$	**$Y_{0}^{600}$ (/pulse)**	$e_{\det}^{600}$
0.1	0.25	0.25	$5\times 10^{-6}$	0.5%

*Note:* The superscript of $Y_{0}$ and $e_{det}$ denotes the aperture of a ground station.

**Table 4 entropy-25-00670-t004:** Sifted key length and secret key length were simulated for the satellite-to-ground QKD with 300 mm/600 mm ground stations for four typical satellite passages. The communication time of Δt corresponds to elevation angles greater than 20°.

Passage	Sifted Key Length, $l_{\mathrm{sift}}$ (kbit)	Secret Key Length, $l_{\sec}$ (kbit)
2021-10-31 ($\theta_{El}^{max}=32.5$°, Δt = 221 s)	547/1207	40/182
2022-03-23 ($\theta_{El}^{max}=42$°, Δt = 260 s)	859/1897	101/357
2022-03-09 ($\theta_{El}^{max}=57.5$°, Δt = 283 s)	1263/2790	187/600
Zenith passage ($\theta_{El}^{max}=90$°, Δt = 285 s)	1586/3505	260/803

## Data Availability

Not applicable.

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
