# Peer review of "Semi-Empirical Satellite-to-Ground Quantum Key Distribution Model for Realistic Receivers"

_entropy, 2023, doi:10.3390/e25040670_

Round 1

Reviewer 1 Report

The authors of the manuscript "Semi-empirical satellite-to-ground quantum key distribution model for realistic receivers" consider a semi-empirical satellite-to-ground QKD model for a more realistic analysis of developed ground stations with 300 and 600 mm telescope apertures. 

According to my opinion this area of research is quite promising. However, the manuscript has some significants flaws. 

In my opinion it is quite odd that there is no significant description of the real ground-based telescopic equipment developed by authors and the experiment conducted for the estimation of the atmospheric extinction coefficients and other model parameters. According to my understanding it is one of the most crucial points of the work compare to the other theoretical models and there is neither detailed expalanation nor reference.

Also, I see no comprehensive comparison with other models, thus no conclusions on the advantages of proposed model can be done. 

Moreover, as I understand authors perform simulation for a given model with adjusted, experimentally measured, parameters. This works well until the analysis section, where only two cases of telescope sizes are considered. Honestly, I did not see any reasoning regarding author’s conclusions. In simple words, the result in its current form may be rephrased as “four times less area of the telescope provides approximately four times fewer key rates”, which sounds obvious. The most interesting question: is this dependence linear? (and I think it is not). In my opinion, in order to make any conclusions there should be equal simulation parameters of the QKD system that do not dependent on the aperture size, and some additional analysis on how the aperture size may affect the dependent ones (e.g. it may affect the background noise or the minimal obstruction of the telescope secondary mirror). Then perform simulations for different aperture sizes, let’s say, from 100 to 1000 mm with a step of 50 or 100 mm. Only significant amount of data points with equal simulation parameters of the QKD system may provide objective answer on the author’s question.

Reviewer 2 Report

Khmelev et al. introduced a novel quantum key distribution model based on experimental determined atmospheric extinction coefficients and analyzed the data from the Micius satellite using this model.

The overall presentation is generally well structured. However, I think the authors should spend more time and effort on the explanation of the figures and theories. It’s hard for the reader to catch the major innovation of the paper. I listed some suggestions below.

1.      I found the explanation of figure 2 incomplete and difficult to understand. For example, how did the author extract the maximum communication time from figure 2?

2.      Can you authors comment on the data shown in table 2?  Why does a portable 300mm telescope have better optical efficiency than a 600mm fixed telescope, while in figure 3, the 300mm one is lossier?

Reviewer 3 Report

Please see the full review attached.

Round 2

Reviewer 1 Report

The authors of the manuscript "Semi-empirical satellite-to-ground quantum key distribution model for realistic receivers" drastically improved the manuscript.

The first concern was clarified. However, absolutely no changes were made in accordance with the second and the most crucial one. Strictly speaking no new model was proposed in the manuscript and the main difference to others is that phenomenological data was utilized. I am not sure if the implementation of the real-experiment parameters to the well-known model should be called the new method as the authors do. Moreover, since no question from the second part of my previous review were solved, this work in its present form seems to me more like technical report rather than scientific paper.

Author Response

Dear reviewer, 

Thank you for taking the time to review our manuscript. We have taken into account your second comment as much as possible in the second revision. As a result, we have done a comprehensive comparison of our semi-empirical method with the model presented by [Vergoossen et al., Acta Astronaut., 2020], which uses the principles set forth by [Bourgoin et al., New J. Phys., 2013].  
Please see page 8 lines 189–210 of the revised manuscript for conclusions on the advantages of the proposed model.

Best regrads,
Authors

Reviewer 2 Report

The authors answered my questions and concerns sufficiently. I recommend accepting in present form.

Author Response

Dear reviewer, 
Thank you for taking the time to assess our manuscript.
Your comments have helped us improve our article.

Best regards 
Authors

Reviewer 3 Report

It is unfortunate to see that the authors did not include all the revisions in the comments, including the suggested references. Please revise according to the comments and send the manuscript for another minor revision.

Round 3

Reviewer 1 Report

Authors have significantly increase the quality of the manuscript and it becomes much more suitable for publication.   

However, I still do not agree with the following point in the conclusion "It has shown that 300 mm station can be successfully employed in satellite-to-ground QKD. Moreover, 300 mm ground-based stations, which can be made much more portable, are shown to be even more cost-effective."   It is quite obvious that the telescope with better characteristics can be also efficient. But what if I'll choose the 300 mm telescope with the same optical characteristics as 600 mm. Will it still be good enough (I think not). Not to mention that no estimation about the cost-efficiency of the telescope is provided.  Honestly, I did not see any reasoning regarding author’s conclusions.  

The only conclusion that can be done here that this specific model of telescope with the following characteristics can be utilized for the following applications. I suggest to rewrite the point since it cannot be generalized, unless the model for dependencies estimation of qkd parameters from the optical parameters of the telecpose is provided.

Author Response

Dear reviewer,
Thank you again for taking the time to assess our manuscript.

You are completely right about the improper sentences in our conclusion. Based on your comments, we changed this section of the conclusion and updated the abstract.

We thank the reviewer for his helpful suggestions and good evaluation of our work.

Best regards,
Authors